# Effects of Changes in Seasonal Weather Patterns on the Subjective Well-Being in Patients with CAD Enrolled in Cardiac Rehabilitation

**DOI:** 10.3390/ijerph19094997

**Published:** 2022-04-20

**Authors:** Dalia Martinaitiene, Nijole Raskauskiene

**Affiliations:** Laboratory of Behavioral Medicine, Neuroscience Institute, Lithuanian University of Health Sciences, Vyduno al 4, LT 00135 Palanga, Lithuania; nijole.raskauskiene@lsmuni.lt

**Keywords:** coronary artery diseases, weather, rehabilitation, subjective well-being, weather sensitivity

## Abstract

Objective: We examined whether seasonal and monthly variations exist in the subjective well-being of weather-sensitive patients with coronary artery disease (CAD) during cardiac rehabilitation. Methods: In this cross-sectional study, 865 patients (30% female, age 60 ± 9) were recruited within 2–3 weeks of treatment for acute coronary syndrome and during cardiac rehabilitation. The patients completed the Palanga self-assessment diary for weather sensitivity (PSAD-WS) daily, for an average of 15.5 days. PSAD-WS is an 11-item (general) three-factor (psychological, cardiac, and physical symptoms) questionnaire used to assess weather sensitivity in CAD patients. Weather data were recorded using the weather station “Vantage Pro2 Plus”. Continuous data were recorded eight times each day for the weather parameters and the averages of the data were linked to the respondents’ same-day diary results. Results: Weather-sensitive (WS) patients were found to be more sensitive to seasonal changes than patients who were not WS, and they were more likely to experience psychological symptoms. August (summer), December (winter), and March (spring) had the highest numbers of cardiac symptoms (all *p* < 0.001). In summary, peaks of symptoms appeared more frequently during the transition from one season to the next. Conclusion: This study extends the knowledge about the impact of atmospheric variables on the general well-being of weather-sensitive CAD patients during cardiac rehabilitation.

## 1. Introduction

Despite significant advancements in prevention and control, cardiovascular diseases (CVDs) continue to be a leading cause of health problems and deaths worldwide [1], including in Lithuania [2]. CVD treatment requires a comprehensive approach due to the significant effects it has on healthcare services. While traditional risk factors such as smoking, alcohol use, hypertension, high cholesterol levels, and obesity are fundamental in explaining a substantial number of CVDs, numerous environmental factors have been discovered to have significant impacts on the risk, development, and severity of CVD [3], for example, the weather. Weather is described as the condition of the atmosphere at any particular time and place. Weather is constantly changing, which partly explains why epidemiologists have conducted little research on the relationships between atmospheric conditions and human health. The prospects for reducing health risks are limited, in contrast to the preventive possibilities that readily flow from the previously mentioned traditional risk factors [4].

Seasonal differences in cardiovascular events, such as acute myocardial infarction, hypertensive urgency, and progressive heart failure have also been well documented and confirmed in large-scale epidemiological studies in various geographic and climatic regions worldwide [5,6,7]. A recent review revealed that seasonality influenced the pattern of nearly every subtype of CVD and that CVD seasonality was most pronounced in individuals living in milder climates, who were the least prepared for extreme weather variations [5]. The effect of season on CVD is complex and, according to the literature, it is modulated by a number of variables. The proposed risk factors include environmental factors such as temperature and exposure to sunlight [8]. Parameters that may increase the risk of acute myocardial infarction include atmospheric variables such as low and high temperature, low atmospheric pressure, high humidity, and reduced exposure to sunlight; the significance of each parameter varies among different seasons [7].

The impact of weather and climate on human health is becoming an increasingly important factor in the context of the current trend towards global and specific regional climatic conditions [9]. Climate change is a gradual change in average weather conditions that changes (mostly increasing) the frequency and intensity of extreme weather events. It is expected that the adverse effects of climate change and the advent of more extreme weather events will become more evident; therefore, the need to find practical ways to reduce the seasonality of CVD [5] and to evaluate the predictors of the development of high weather sensitivity to avoid the risk of the complications provoked by the unfavorable weather conditions [10] has been emphasized.

Most studies on the effects of seasonality on CVD have been performed retrospectively and, usually, hospital or emergency aid services databases have been used to identify acute events associated with CVD. Some authors have suggested that well-conducted studies of limited geographical areas with clearly defined parameters of interest and specific risk groups were better than retrospective analyses of large datasets that, so far, have provided confusing data [3,11]. Rehabilitation is one of the main aspects included in CVD medical care. Cardiac rehabilitation programs aim to reduce the risk of another heart event, to monitor and control the current heart condition, and to improve the health and quality of life of patients with CVDs [12]. Cross-sectional studies conducted during rehabilitation programs have also provided additional evidence of a link between weather conditions and health and/or well-being, as well as provided more information to health professionals and the general public. Awareness about the potential influence of weather conditions on the well-being of patients with heart disease may contribute to the planning and implementation of actions leading to improved medical care services and preventative measures that help to avoid worsening of well-being in the future.

Previously, we conducted a study aimed at evaluating the association between the subjective well-being of patients with coronary artery disease (CAD) and daily weather parameters [13]. We conducted a weather sensitivity survey in Lithuania, a country in northeastern Europe with four different seasons and different winter–summer conditions. Using the same data, in this study, we aim to determine if seasonal and monthly variations exist in the subjective well-being of weather-sensitive CAD patients during cardiac rehabilitation and if this variation is related to meteorological parameters.

## 2. Materials and Methods

### 2.1. Sample and Procedure

A detailed description of the study methodologies is presented in our recent study [13]. We used the STROBE cross sectional checklist when writing the report for this cross-sectional study [14]. Briefly, we enrolled 865 patients with CAD attending a rehabilitation program at the Palanga Clinic of the Lithuanian University of Health Sciences Neuroscience Institute (LUHS NI) from June 2008 to October 2012 (Figure 1). Inclusion criteria included an established diagnosis of CAD and age of 18 years or older. Patients were excluded from the study if they had undergone coronary artery bypass graft surgery, cognitive disorientation, communicative disabilities, or other severe diseases, or did not speak Lithuanian fluently. Patients were referred to the rehabilitation clinic within two weeks of treatment for acute coronary syndromes. The duration of cardiac rehabilitation varied based on the diagnosis, from 14 to 20 days. All patients got standard secondary prevention of CAD treatment according to the existing guidelines, including cardiologists’ prescribed medication, physical therapy, risk factor management, and nutritional and psychological counseling. Moreover, physical therapy is also comprised of daily 1.5-h outside walking sessions. Within three days of admission to the rehabilitation clinic, patients were assessed for demographics (age and gender) and clinical characteristics (including NYHA functional class).

### 2.2. Well-Being

To evaluate well-being, all patients completed the Palanga self-assessment diary for weather sensitivity (PSAD-WS). PSAD-WS is a valid and reliable 11-item (general) three-factor questionnaire for collecting information regarding weather sensitivity in patients with CAD [15]. PSAD-WS was created as a self-assessment diary consisting of a list of symptoms. Questionnaire validation was performed on the same sample of patients with CAD. The three factors or symptom subscales reflected: (1) psychological symptoms (apathy, indolence, lack of energy, weakness, daytime sleepiness, and bad mood); (2) cardiac symptoms (palpitation, irregular heartbeat, heartache, and stabbing pain in the chest area); and (3) physical symptoms (joint pain and numbness in the extremities). There were four different PSAD-WS scores defined: a score concerning psychological symptoms, a score concerning cardiac symptoms, a score concerning physical symptoms, and a total score. The total score was the sum of all 11 items [15]. Each day, for a period from 8 to 21 days, depending on the length of the rehabilitation program (average 15.5 ± 3.1 days), all patients (*n* = 865) completed the PSAD-WS questionnaire about their well-being on the last day, including the nighttime. They were only required to indicate whether one or more of the symptoms listed in the self-assessment diary existed on the day of completing the questionnaire. We did not ask patients to relate their well-being to weather conditions. The severity of symptoms was rated according to 0 (not expressed at all), 1 (expressed), or 2 (strongly expressed). During the study period (between June 2008 and October 2012), a total of 13,327 well-being measurements were taken.

### 2.3. Weather Sensitivity

We asked patients the following question to assess their self-perceived weather sensitivity: “Do you feel the weather changes?” The possible answers were “NO” or “YES”. When patients replied “YES”, they were classified as weather sensitive (WS).

### 2.4. Study Location and Weather Data

During the study period, the weather variable observations were carried out using a “Vantage Pro2 Plus” weather station, located on the roof of the Palanga Clinic of the LUHS NI (in the same location where the respondents attended the rehabilitation program). Palanga city is located in northwest Lithuanian on the eastern shore of the Baltic Sea and belongs to the Lithuanian coastal climate region (55°58′ N, 21°03′ E). The coastal climate region belongs to the Baltic coastal climate region, with climatic indexes that are the most different from the other three Lithuanian climate regions and are closer to Europe’s marine northwest climate.

Eight times each day with three-hour intervals (at 0, 3, 6, 9, 12, 15, 18, and 21 h), continuous data of the weather parameters (atmospheric pressure in hectopascal, the temperature in Celsius, relative humidity in percentage, wind speed in meters per second, wind direction in degrees (in a clockwise direction from true north (0 to 360°)), and solar radiation in watt per square meter) were recorded and automatically transmitted to the database using sensors by radio, and then stored in a matrix form. The averages of the weather parameter data recorded eight times daily were calculated and linked to the patients’ same-day self-assessment diary responses. The results referred to more than four years and included all seasons. The classification of seasons was based on the dates when the patients completed the self-assessment diaries. According to the meteorological season calendar, spring begins on 1 March, summer on 1 June, autumn on 1 September, and winter on 1 December.

### 2.5. Statistical Analysis

Demographic variables and clinical conditions were summarized by descriptive analyses. For descriptive purpose, the participants were grouped based on the presence or absence of weather sensitivity (self-reported), and then compared with clinical and demographic variables, using χ^2^ tests and odds ratio (OR). The correlations between the PSAD-WS subscales and weather parameters were assessed using the Spearman correlation coefficient. Line graphs were used to present the distributions visually. ANOVA was conducted to examine if different patient characteristics (reported weather sensitivity, age group, gender, and NYHA functional class) had overall effects on PSAD-WS concepts (discriminate validity). The effects are reported as an F statistic and its associated degrees of freedom and *p*-value.

The following form of statistical analysis was used: days and the expression of patients’ well-being in those days were taken as the units of analysis. Daily scores of PSAD-WS were aggregated from records on individual patients at current day. The results of our previous study [13] showed that the patients who described themselves as being weather-sensitive had almost two times greater possibility to fill above the median on PSAD-WS than patients who described themselves as non-weather-sensitive. The main outcome variable used in the analysis was the cardiac symptoms PSAD-WS subscale. Four multiple regression models (forward stepwise method) were developed for the seasons to predict patient well-being according to the cardiac symptom subscale. The independent variables for all models were: gender; age; interaction (gender × temperature); psychological symptoms subscale; physical symptoms subscale; and the daily mean of atmospheric pressure, temperature, relative humidity, and solar radiation. The results from the regression analyses were reported as beta coefficients and R^2^.

Statistical analyses were performed using the SPSS Statistical Software (version 17.0, SPSS Inc., Chicago, IL, USA); a *p*-value of less than 0.05 was considered to be statistically significant.

## 3. Results

A total of 865 patients with CAD were enrolled in the study (Figure 1): 609 (70%) males and 256 (30%) females with a mean age of 60 ± 9 years, and a range from 32 to 88 years. Almost half of the patients’ responses to the question “Do you feel the weather changes?”) indicated that they were WS (*n* = 410, 47.3%).

Table 1 shows the demographic and clinical characteristics of all study participants, stratified by self-reported weather sensitivity. Female patients (OR = 2 and 95% CI 1.7–3.0) and patients over 60, adjusted for gender (OR = 1.6 and 95% CI 1.5–1.7) were more likely to identify as WS. The possibility of being WS increased with a higher NYHA functional class.

The wording of a single question can introduce substantial differences and may emphasize only one dimension of weather sensitivity. Within our study, we chose a multidimensional assessment of weather sensitivity. There was a significant effect (Table 2) of patient’s weather sensitivity type (no or yes) on all PSAD-WS symptom subscale scores. There was a significant effect of age group (*p* = 0.020) and gender (*p* < 0.001) only on the physical symptoms score. The NYHA functional class affected physical symptoms (*p* = 0.001) and psychological symptoms (p < 0.001) scores, but not cardiac symptom (*p* = 0.250) scores.

The effects of weather parameters on the PSAD-WS total score and symptom subscale scores were found to be weak. Meanwhile, the analysis of the female group revealed only one significant association between temperature and cardiac symptoms. Furthermore, there was interaction (gender x temperature). The associations between temperature and cardiac symptoms differed between male and female patients: at higher temperatures, there were more cardiac symptoms in female patients, whereas at lower temperatures, there were more symptoms in male patients (Table 3).

The mean value distributions of temperature, solar radiation, relative humidity, and barometric pressure, according to month, during the study period, are shown in Figure 2.

The percentages of measurements per month, of patients with CAD who reported one or more symptoms on the PSAD-WS total scale and symptom subscales were visualized according to reported WS (Figure 3). There were more reported symptoms in the group describing themselves as WS as compared with patients describing themselves as not WS on the PSAD-WS total scale and in three of the symptom subscales. In addition, the percentage of measurements with symptoms fluctuated throughout the year, and this fluctuation was more pronounced in the WS group. The WS group clearly showed expressed peaks, which were more pronounced in the separate symptom groups. The highest incidences of symptoms were reported in August (summer), December (winter), and March (spring); the lowest incidences of symptoms were reported in January, July, and November (PSAD-WS total scale).

Analyses of individual symptom groups showed that WS patients were dominated by a wide range of psychological symptoms (40–80% of measurements per month), with pronounced peaks in August (summer), April (spring), and October (autumn). Cardiac symptoms (30–60% of measurements per month) were more pronounced in March (spring), August (summer), and December (winter); physical symptoms (30–60% of measurements per month) were more pronounced in March and May (spring) and September (autumn). The data show that peaks in symptoms, especially cardiac symptoms, are more evident during the transition from one season to the next. Looking at the weather data during the study period, we can observe more pronounced changes in weather parameters during the peaks of symptoms, i.e., March to April, increase in solar radiation, increase in atmospheric pressure, and significant decrease in humidity (Figure 2) and August, rapid decrease in solar radiation and increase in humidity.

### Seasonal Associations between Subjective Well-Being and Weather Parameters: Multivariate Approach (among WS Patients)

Analyses were performed only among patients who described themselves as WS (*n* = 410, 6361 days of well-being measurements, average 15.3 day for each patient). As different symptoms groups showed different seasonal distribution; in this case, we paid main attention only to cardiac symptoms (outcome) (Table 4).

In spring, the month with the highest number of cardiac symptoms reported by WS patients was March, whereas the month with the lowest number was May (*p* < 0.001) (Table 4). The regression analysis revealed a significant interaction between gender and temperature. The reported cardiac symptoms were independently associated with lower NYHA functional class, higher temperature (for female patients (interaction)), and lower solar radiation.

In summer, the highest number of cardiac symptoms were reported in August and the lowest number were reported in June (*p* < 0.001). More cardiac symptoms were reported by female patients, by younger than 60-year-old patients with a higher NYHA functional class. Lower atmospheric pressure was an independent factor for the presence of cardiac symptoms. Lowering atmospheric pressure was found to be an independent predictor of the presence of cardiac symptoms.

In autumn, the lowest number of cardiac symptoms were reported in October, whereas the highest number of cardiac symptoms were reported in September (*p* < 0.001). More cardiac symptoms were reported by female patients and by patients with lower NYHA functional class, and a higher number of symptoms were associated with lower atmospheric pressure.

In winter, the highest number of cardiac symptoms were reported in December, and the lowest number of cardiac symptoms were reported in January (*p* < 0.001). For male and female patients, the chance of having cardiac symptoms in the cold period was the same. It was associated with patients older than 60 years of age and higher NYHA functional class with the presence of psychological symptoms.

## 4. Discussion

The purpose of this study was to determine if seasonal and monthly variations existed in the subjective well-being of WS patients with CAD during cardiac rehabilitation and if this variation was related to meteorological parameters. We found that WS patients reported being more sensitive to seasonal changes than patients who were not WS. The highest number of symptoms were reported in August (summer), December (winter), and March (spring) (PSAD-WS total scale and cardiac symptoms subscale), suggesting that the peak of symptoms was more pronounced during the transition from one season to the next, when the fluctuations in weather parameters were more pronounced. Meanwhile, the peaks in the different symptom groups were unequal, and psychological symptoms were most common. Peaks in psychological symptoms appeared in August, April, and October, and most of the physical symptoms were reported by patients who were WS in March, May, and September. It should be noted that Lithuania is a country in northeastern Europe with four different seasons and different winter–summer conditions; our study was performed in the Lithuanian coastal climate region, which is characterized by a cool spring and cool summer; a warm, often without permanent snow cover in winter; a warm and rainy autumn; and small fluctuations in daily and annual temperature [16].

We were unable to find studies that examined the subjective well-being of patients with CAD during the rehabilitation period; therefore, direct comparisons with previous studies are limited. The findings of this study on subjective well-being are consistent with previous results from large-scale epidemiological studies conducted in various geographic and climatic regions worldwide, i.e., CVDs are affected by seasonal variations [5,6,8,17,18]. However, the seasonal variations do not appear to be universal. Most studies have indicated that the winter season had the highest rate of CVD-related hospitalizations and mortality [17,18,19,20] and that event rates were typically 10–20 percent higher than in the summer [5]. It has been suggested that seasonality in CVD was particularly prominent in people who live in milder climates and, therefore, are less prepared for extreme weather changes [5]. Contrary to the general misconception, the majority of temperature-related deaths occur at milder, non-optimal temperatures [19]. In a study in the Czech Republic, excess deaths due to ischemic heart disease (IHD) during hot spells were found to be mainly among people with chronic illnesses whose health had already been compromised. Meanwhile, cardiovascular changes induced by cold stress may result in deaths from acute coronary events rather than chronic IHD [21].

The seasonality of CVD is likely to be caused by the complex interplay between human vulnerability and environmental conditions [5]. Some potential risk factors suggested by researchers are temperature, sunlight exposure, air pollution, some characteristics of the population (age, sex, location, and socioeconomic status), lifestyle (physical activity, smoking, and eating habits), infections, hormones, and vitamin D [9,17]. Weather sensitivities are not always straightforward and are often identified as an accelerator or catalyst rather than a causal factor. In a large nationwide study in Sweden, low air temperature, low atmospheric air pressure, high wind velocity, and shorter sunshine duration were associated with the risk of heart attacks, with the most evident association observed for air temperature [22]. Another nationwide study in the Republic of Slovenia, a South-Central European country [23], found that daily average temperature, atmospheric pressure, and relative humidity had relevant and significant influences on the incidence of acute coronary syndrome for the entire population. In a study performed in Lithuania [24], some patterns of relative humidity, cloud cover, and daily changes in atmospheric pressure and relative humidity were associated with the risk of some types of strokes.

Our study revealed that subjective cardiac symptoms were independently associated with lower atmospheric pressure in summer and autumn and lower solar radiation in spring. In contrast to previous studies, which have shown that increased morbidity and mortality for CVD were most associated with temperature, both high and low [1,25], we found only one significant association with temperature and only in spring. This might be explained by the fact that study participants were already recovering from an acute event, were enrolled in a rehabilitation program (which included care, medicines, a different diet, and a different daily routine than at home), and spent more time inside. Meteorological changes are also thought to impact other risk factors of CAD control. For instance, cold days may limit physical activity and promote excessive or high-calorie food intake, alcohol consumption, smoking, etc., which may also contribute to the seasonality of symptoms. Furthermore, these observations support those of previous studies reporting that the significance of each weather parameter varied among different seasons. In a study performed in Italy, a higher risk of primary percutaneous coronary intervention was found with lower minimum atmospheric pressure in the preceding days, lower rainfall in winter, greater changes in atmospheric pressure in spring, and higher temperatures in summer [26].

In most cases, the physiological responses that occur at a specific temperature, humidity, sunlight, and other values or changes in those values are well understood [6,7,27]. Non-optimal temperatures have a variety of effects on humans’ physiological systems, as well as interact with pre-existing diseases and chronic disorders. Even when body temperature remains normal, thermoregulation strains the cardiovascular system [28]. The higher or lower the temperature and the longer the exposure, the more work is required of the cardiovascular system to maintain an optimal temperature. Present theories suggest that lower atmospheric pressure, which affects the sympathetic nervous system and the immune system, increases blood pressure, and that changes in atmospheric pressure can affect atherosclerotic plaques, which cause the plaques to rupture [29,30].

As in previous studies [5,9,23], our study showed that seasonal differences between age and gender groups were unequal. Almost in all seasons, females experienced more cardiac symptoms than males, meanwhile, in winter, the risk of having cardiac symptoms for males and females was the same. Furthermore, the only one association with temperature in spring had a significant gender interaction, suggesting that, in spring, females were more likely to experience cardiac symptoms at higher temperatures and males were more likely to experience symptoms at lower temperatures. Gender differences have also been found in previous weather-health related studies [27,31,32], which were assumed to be due to the different thermoregulatory systems, different physiological structures of males and females (differences in body fatness and distribution, body surface area, and mass), and the varied impacts of bioclimatic conditions on neurohormonal systems [31,32,33]. We found that, in summer, more cardiac symptoms were associated with patients who were younger than 60 years old, meanwhile, in winter, cardiac symptoms were associated with patients who were older than 60 years old. This is consistent with other studies’ results showing that, during a cold period, the risk of acute coronary events [34], acute myocardial infarction [23], and CHD mortality [35] increased with age.

Our study showed that WS patients were more likely to experience psychological symptoms during cardiac rehabilitation. Psychological factors are known to affect the biological processes associated with CAD progression [36]. Data suggests that the incidence of depression in people with heart failure is 20% higher than in healthy people [37]. Psychologically distressed people or people with sensitive nervous systems are another well-known WS group [38]. Although psychological symptoms were not thoroughly examined in this study, it can be assumed that environmental factors would contribute more significantly to the seasonal well-being of patients during cardiac rehabilitation if individuals experience significant mental distress symptoms.

Our results extend the knowledge about the impact of atmospheric variables on humans’ health and well-being. Even though we analyzed the subjective well-being of patients during rehabilitation, the general findings of our study were in line with the findings of the majority of studies, which have also noted seasonality in CVD and associations with weather parameters such as atmospheric pressure, solar radiation, and temperature. The seasonality of the perceived symptoms may indicate that environmental factors are involved in cardiovascular diseases. Seasonal variations in various symptoms may aid in interpreting the association between the symptoms and other factors in cardiac rehabilitation across different seasons. Knowing about patients’ weather sensitivity can help physicians to understand and interpret patients’ complaints. If patients describe themselves as weather sensitive, physicians should be aware that these patients will be more responsive to atmospheric conditions, especially with changing seasons. If individuals experience significant mental distress symptoms, environmental factors may contribute more to the seasonal well-being of patients during cardiac rehabilitation.

### Strength and Limitations

The present study involved a relatively large sample size of CAD patients who completed their rehabilitation in the same clinic under the same conditions. The results spanned four years and included all seasons. Separate monthly and seasonal analyses were performed. Despite these strengths, there were some limitations. We used only available outdoor atmospheric parameter data to evaluate the impact of climatic variables on the well-being of patients and did not assess the time that patients spent outside. The effect of climatic factors on various well-being symptoms could have been moderated by indoor conditions and time spent outdoors. Some risk factors, such as socioeconomic status and lifestyle, were not taken into consideration.

## 5. Conclusions

Our results confirm the importance of seasonal atmospheric state variability in the general well-being of weather-sensitive CAD patients during cardiac rehabilitation. Seasonal variations in some cardiac, psychological, or physical symptoms could affect patients undergoing cardiac rehabilitation, especially if patients describe themselves as weather sensitive. This pattern changes slightly according to age and gender. If patients experience significant mental distress symptoms, their sensitivity to environmental factors will be higher. This study extends the knowledge about the impact of atmospheric variables on the general well-being of weather-sensitive CAD patients during cardiac rehabilitation.

## Figures and Tables

**Figure 1 ijerph-19-04997-f001:**
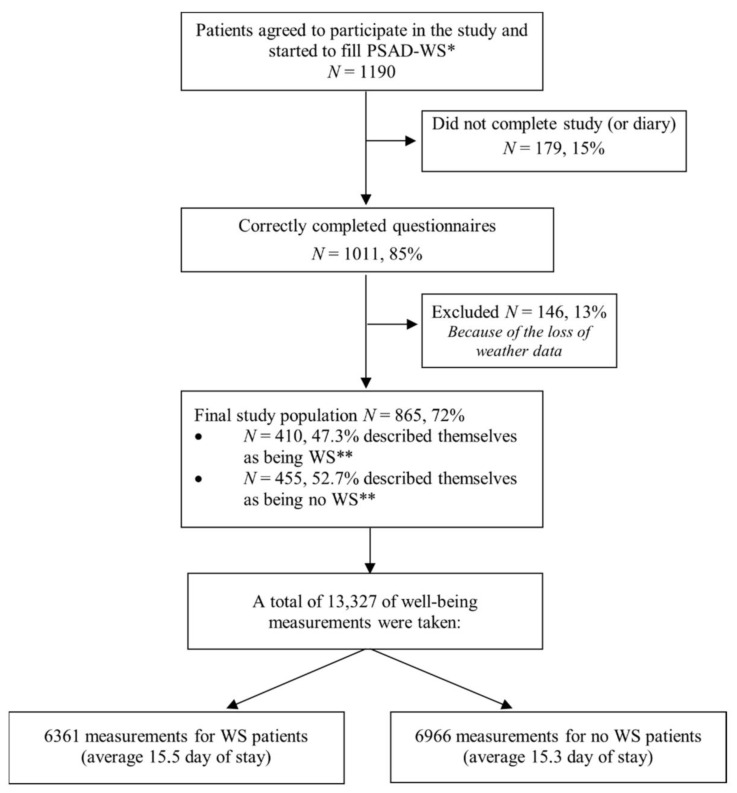
Flowchart of the study. * PSAD-WS, Palanga self-assessment diary for weather sensitivity; ** WS, weather sensitive.

**Figure 2 ijerph-19-04997-f002:**
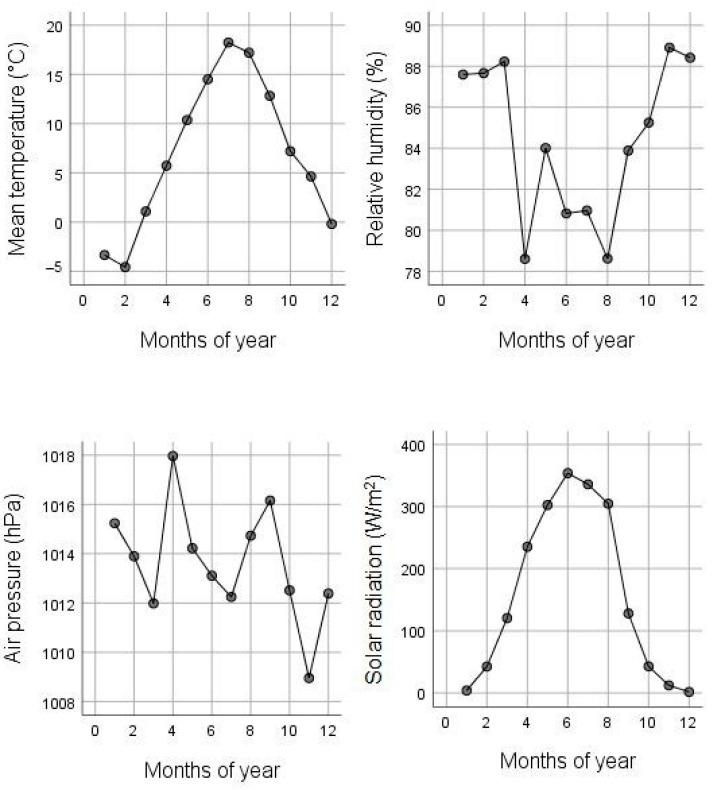
Mean values of weather variables according to month, during the study period (from June 2008 to October 2012). Error bars denote a 95% interval.

**Figure 3 ijerph-19-04997-f003:**
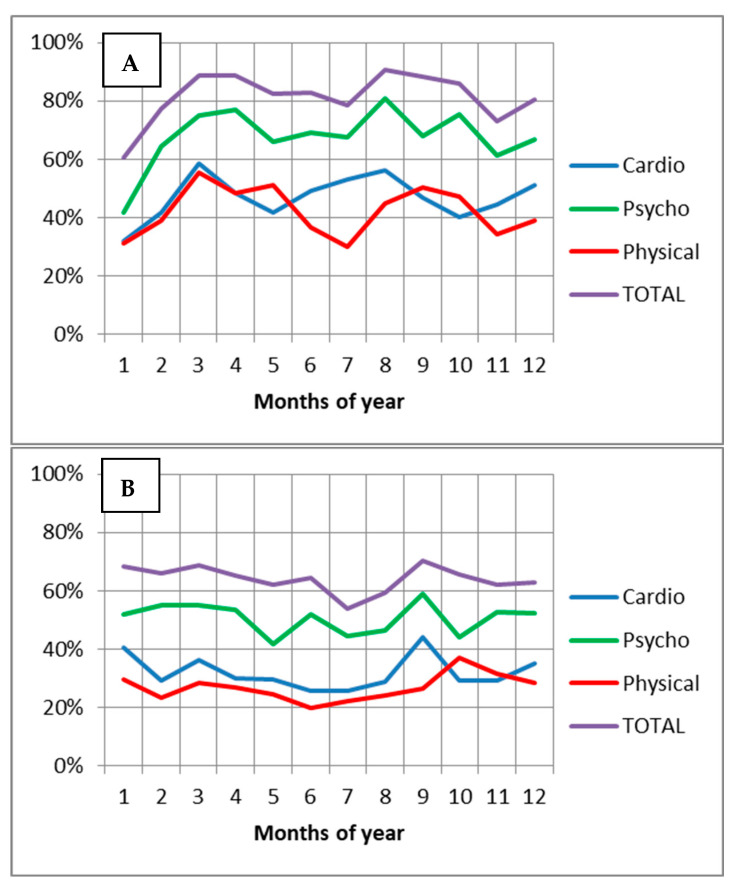
Percentages of measurements per month, of patients with CAD who reported one or more subjective well-being symptoms on the PSAD-WS total scale and symptom subscales: (**A**) Weather sensitive patients; (**B**) not weather sensitive patients.

**Table 1 ijerph-19-04997-t001:** Demographic and clinical characteristics of all patients at inclusion and stratified by self-report weather sensitivity.

Variable	*n* (%)	Not Weather Sensitive *n* = 455	Weather Sensitive *n* = 410	Odds Ratio (95% CI)
Age, years:				*p*^for trend^ < 0.01
<50	133 (15.4)	85 (18.7)	48 (11.7)	1
51–60	280 (32.4)	161 (35.5)	119 (29)	1.3 (0.8–1.9)
61–70	324 (37.4)	147 (32.2)	177 (43.2)	2.1 (1.4–3.2)
>70	128 (14.8)	62 (13.6)	66 (16.1)	1.9 (1.1–3.1)
Gender:				*p*^for trend^ < 0.001
Male	609(70)	356 (78.3)	243 (61.7)	1
Female	256 (30)	99 (21.7)	157 (38.3)	2.2 (1.7–3.0)
NYHA class:				*p*^for trend^ < 0.001
I	54 (6.2)	44 (9.7)	10 (2.4)	1
II	552 (63.9)	301 (66.1)	251 (61.2)	3.6 (1.8–7.4)
III	259 (29.9)	110 (24.2)	149 (36.4)	5.9 (2.8–12.2)

NYHA, New York Heart Association.

**Table 2 ijerph-19-04997-t002:** Analysis of variance by PSAD-WS subscales (F statistic).

	PSAD-WS Subscales		
	Psychological Symptoms	Cardiac Symptoms	Physical Symptoms
Weather sensitive (no vs. yes)	F(1,864) = 27.1 *p* < 0.001	F(1,864) = 6.3 *p* = 0.012	F(1,864) = 30 *p* < 0.001
Age groups <50, 51–60, 61–70, 70+	F(3,862) = 2.2 *p* = 0.093	F(3,862) = 1.3 *p* = 0.27	F(3,862) = 3.3 *p* = 0.020
Gender, female vs. male	F(1,864) = 2.2 *p* = 0.139	F(1,864) = 3.3 *p* = 0.070	F(1,864) = 21.8 *p* < 0.001
NYHA class	F(3,864) = 12 *p* < 0.001	F(3,864) = 1.37 *p* = 0.25	F(3,864) = 5.6 *p* = 0.001

**Table 3 ijerph-19-04997-t003:** Spearman correlation coefficients (r) between weather parameters and daily PSAD-WS, according to gender.

	Atmospheric Pressure hPa	Temperature °C	Relative Humidity %	Solar Radiation W/m^2^
Males				
PSAD-WS total	0.005	−0.043 **	0.043 **	−0.042 **
Psychological symptoms	−0.009	−0.036 **	0.042 **	−0.029 *
Cardiac symptoms	0.014	−0.024 *	0.017	−0.026 *
Physical symptoms	0.025 *	−0.051 **	0.041 **	−0.056 **
Females				
PSAD-WS total	0.005	0.026	−0.024	−0.027
Psychological symptoms	0.014	0.019	−0.027	−0.027
Cardiac symptoms	−0.019	0.031 *	−0.004	−0.023
Physical symptoms	0.008	0.007	0.008	−0.007

* *p* < 0.05 and ** *p* < 0.01.

**Table 4 ijerph-19-04997-t004:** Multiple linear regression analysis predicting the reporting cardiac symptoms scores separately by four seasons (significant standardized regression coefficients β).

	Dependent Variable Sum of Cardiac Symptoms			
Independent Variables	Spring	Summer	Autumn	Winter
Gender (1 = M; 2 = W)		0.104	0.171	
Interaction: Gender (1 = M; 2 = W) × (temperature)	0.107			
Age ≥60 vs. <60 years		−0.057		0.072
NYHA class	−0.085	0.169	−0.249	0.091
Solar radiation	−0.064			
Atmospheric pressure		−0.082	−0.089	
May vs. March, April	−0.124			
August vs. June, July		0.058		
October vs. September, November			−0.096	
December vs. January, February				0.167
Sum psychological symptoms subscale	0.379	0.285	0.369	0.544
Sum physical symptoms subscale	0.199	0.291	0.201	
Model	R^2^ = 0.266 F = 93.73 *p* < 0.001	R^2^ = 0.287 F = 71.01 *p* < 0.001	R^2^ = 0.242 F = 74.51 *p* < 0.001	R^2^ = 0.369 F = 171.34 *p* < 0.001

Number of subjects included in the analysis *n* = 410 WS patients, 6361 measurements.

## Data Availability

The data that support the findings of this study are available from the corresponding author upon reasonable request.

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
