# Peer review of "Effects of Changes in Seasonal Weather Patterns on the Subjective Well-Being in Patients with CAD Enrolled in Cardiac Rehabilitation"

_ijerph, 2022, doi:10.3390/ijerph19094997_

Round 1
Reviewer 1 Report
I believe this paper is a valuable study that investigates whether there is seasonal or monthly variation in subjective well-being during cardiac rehabilitation in weather-sensitive coronary artery disease (CAD) patients. I think "Disscussion" is also very aptly described. Upon publication, I would suggest two minor modifications.
Materials and Methods 2.5. Statistical analysis L158
With respect to statistical analysis, the stepwise method is said to be prone to bias in parameter estimation (Whittingham et al. 2006. Why do we still use stepwise modelling in ecology and behaviour? Journal of Animal Ecology 75:1182-1189.). For this reason, we prefer not to use the stepwise method in variable selection. If the stepwise method is still used, I think it is necessary to add a description of how the bias was taken into account.
3. Results (L185-186)
”NYHA class affected Physical symptoms (p=0.001) and Psychological symptoms scores (p<0.001), but not on Cardiac symptoms scores (p=0.250).”
Since this is the result of a cross-sectional study, it is more accurate to say "related" than "affected”.
Reviewer 2 Report
Your paper describes the relationship between patient questionnaire derived symptoms (psychological, cardiovascular, neuro musculoskeletal) and seasonal variations in selected components of weather in Lithuania (with 4 distinct seasons). Your declared main outcome measure was cardiovascular symptoms.
I don't think that the title accurately reflects the focus of your paper. I suggest consideration of changing the title to, " Effects of changes in seasonal weather patterns on symptoms in patients enrolled in cardiac rehabilitation".
I do not believe English is your primary language and the writing style and word usage needs to be improved dramatically. I suggest having an individual who is fluent in scientific English carefully edit the paper.
I suggest that you briefly describe the cardiac rehabilitation program. I assume it is a standard residential rehabilitation program common in Lithuania. It would be helpful to know if patients spend time outdoors year around as part of the rehabilitation experience.
I suggest that you explain your definition of "weather sensitive". Your definition is: "Do you feel the weather changes?" To me, this is very superficial. Do you mean that patients with weather sensitivity experience worsening of their symptoms, especially cardiovascular symptoms, during changeable weather?
The Results section is overly complicated and should be reduced in length by at least 1/3. I would suggest that you focus on your main findings, emphasizing cardiovascular symptoms (your main outcome measure). I would suggest that you provide the frequency and severity of each cardiovascular symptom that was weather sensitive. I would also suggest that you discuss whether or not weather sensitive cardiovascular symptoms resulted in changes in the medical care of the patients (improved patient care).
Reviewer 3 Report
Dear Editor,
Thank you for in this very interesting topic. Here are my comments.
- I would recommend a title change. The title implies that there are seasons for cardiac rehab. Also, there is only a very weak relationship here between the study and the cardiac rehab program. Finally, the cohort does not include coronary artery bypass patients, which is a very large segment of the rehab population. It seems like the focus here is on weather sensitivity and reporting of symptoms.
- The description of cardiac and physical symptoms should be clarified. What is meant by heartache? Is this angina pectoris? There are other descriptors of angina besides stabbing. Were these considered. Also, women tend to have different descriptions of pain, such as shortness of breath. How were dyspnea, syncope, edema, claudication, etc. considered?
- What is the validity of the PSAD-WS? It seems like more detail is required to determine if someone is truly WS, or is it a subjective assessment?
- Was a power analysis done to ensure that the sample size was large enough to answer the research question?
- Why are only WS positive patients analyzed? Patients that are WS negative still report symptoms.
- Tables 4-6 sum the symptoms. Is there any consideration for the quality or severity of the symptom? You could argue that stabbing chest pain is more important than feeling a palpitation.
- The timing of the reported symptom compared to the start of the program might be an important consideration (or a confounder). How long were patients monitored? If they were monitored in cardiac rehab, was that a 12 week period? If so, was there any seasonality to beginning the program (i.e., more patients started in the winter, when there are more cardiac events, and fewer in the warmer months), and could that impact the analysis?
- Finally, a lot of data is presented, but I am still confused as to the relationship between symptom reporting and atmospheric conditions.
Round 2
Reviewer 2 Report
Your revised paper is improved. Congratulations on your nice project.
Author Response
Thank you very much!
Reviewer 3 Report
Dear Authors,
Thank you for your thoughtful comments and revisions. I believe you should incorporate your comments in more detail into the manuscript because I think many readers will have the same questions.
Author Response
Author’s response to Reviewer3
Title: Which season is better for cardiac rehabilitation?
International Journal of Environmental Research and Public Health; ID: ijerph-1622370
Authors: Dalia Martinaitiene, Nijole Raskauskiene
Version: 2
Date: 7 April 2022
Dear Reviewer,
Thank you again for your comments concerning our manuscript entitled “Which season is better for cardiac rehabilitation?” (ID: ijerph-1622370) by Dalia Martinaitiene and Nijole Raskauskiene. We appreciate the time and effort that you have dedicated to providing your valuable feedback on our manuscript.
Here is a response to your comments.
Response to Reviewer 3 Comments
Dear Authors,
Thank you for your thoughtful comments and revisions. I believe you should incorporate your comments in more detail into the manuscript because I think many readers will have the same questions..
Response: Thanks for your comment. We agree with you and apologize for not fully understanding your recommendations. We supplemented section 2.2. Well-Being and 2.5 Statistical Analysis in Methods part following (in red). Also, we would like to notice, that the English language of our article was edited by MDPI.
“2.2. Well-Being
To evaluate well-being, all patients completed the Palanga self-assessment diary for weather sensitivity (PSAD-WS). PSAD-WS is a valid and reliable 11-item (general) three-factor questionnaire for collecting information regarding weather sensitivity in patients with CAD [15]. PSAD-WS was created as a self-assessment diary consisting of a list of symptoms. Questionnaire validation was performed on the same sample of patients with CAD.
2.5. Statistical Analysis
The following form of statistical analysis was used: days and the expression of patients' well-being in those days were taken as the units of analysis. Daily scores of PSAD-WS were aggregated from records on individual patients at current day. The results of our previous study [13] showed that the patients who described themselves as being weather-sensitive had almost 2 times greater possibility to fill above the median on PSAD-WS than patients who described themselves as non-weather-sensitive“.
Sincerely,
Dalia Martinaitiene, PhD
Laboratory of Behavioral Medicine,
Neuroscience Institute, Lithuanian University of Health Sciences
Vyduno 4, Palanga, LT-00135, Lithuania.